# Maternal Factors and the Utilization of Maternal Care Services Associated with Infant Feeding Practices among Mothers in Rural Southern Nepal

**DOI:** 10.3390/ijerph16111887

**Published:** 2019-05-28

**Authors:** Dilaram Acharya, Jitendra Kumar Singh, Rajendra Kandel, Ji-Hyuk Park, Seok-Ju Yoo, Kwan Lee

**Affiliations:** 1Department of Preventive Medicine, College of Medicine, Dongguk University, Gyeongju 38066, Korea; dilaramacharya123@gmail.com (D.A.); skeyd@naver.com (J.-H.P.); medhippo@hanmail.net (S.-J.Y.); 2Department of Community Medicine, Kathmandu University, Devdaha Medical College and Research Institute, Rupandehi 32907, Nepal; 3Department of Community Medicine and Public Health, Janaki Medical College, Tribhuvan University, Janakpur 44618, Nepal; jsingdj@gmail.com; 4Personal Social Services Research Unit, London School of Economics and Political Science, London WC2A2AE, UK; r.kadel@lse.ac.uk

**Keywords:** maternal care services, infant feeding practices, maternal factors, MATRI-SUMAN, Nepal

## Abstract

This study aimed to investigate the maternal factors and utilization of maternal care services associated with infant feeding practices in rural areas of Southern Nepal. Data from a cluster randomized controlled trial ‘MATRI-SUMAN’(Maternal Alliance for Technological Research Initiative on Service Utilization and Maternal Nutrition) conducted between 2015–2016 were analyzed. A total of 426 pregnant women in their second trimester were recruited from the MATRI-SUMAN trial, which was conducted on six villages in rural areas of the Dhanusha district, Nepal. A total of 379 mothers that had ever breastfed their infants, and followed for at least seven months after birth were included in the analysis. Multivariate logistic regression analysis was used to identify independent risk factors associated with child feeding practices after controlling for potential confounders. Of the 379 mothers, 41.4%, 53%, and 43% initiated breast feeding within the first hour of birth (EIBF), practiced exclusive breastfeeding (EBF), and initiated timely complementary feeding (CF) at six months, respectively. Multiple logistic regression results revealed that maternal education (secondary or higher), an occupation in the service/business/household sectors, receipt of MATRI-SUMAN intervention, more than four ANC (antenatal care) visits, and delivery in a health facility were associated with higher odds ratios of EIBF. Similarly, mothers with a primary, secondary and higher level of education, that worked in the service/business/household sectors, primiparous mothers, those that received MATRI-SUMAN intervention, visited ANC more than four times, and made a PNC (postnatal care) visit had higher odds ratios of EBF, while mothers who were 35–45 years of age were less likely to have used EBF. In addition, education to the secondary or a higher level, a male baby, receipt of MATRI-SUMAN intervention and a PNC visit had higher odds ratios of CF initiation at six months. The promotion of maternal ANC visits, birth at a health institution, and postnatal visits should be recommended in order to improve child feeding practices in Nepal.

## 1. Introduction

Despite the substantial progress made to reduce child mortality, 5.6 million children aged less than five years died worldwide in 2016, and the majority of these deaths occurred within the first year of life [1]. Furthermore, it was estimated in 2011 that malnutrition was responsible 3.1 million of these deaths worldwide [2]. Appropriate child feeding practices include optimal breast feeding and complementary feeding, and are key determinants of child nutritional wellbeing and health [2,3,4]. The large bodies of literature support the notion that the initiation of breast feeding within the first hour of birth (EIBF) protects children from common childhood diseases, such as diarrhoea and acute respiratory infections (ARI) [5,6,7], and can significantly reduce neonatal mortality [8,9]. Exclusive breast feeding (EBF) also decreases the incidences of diarrhoea and ARI [7,10] and prevents growth faltering and acute malnutrition [11,12]. Furthermore, recent systematic reviews and meta-analyses have reported that children who are breastfed for longer periods have fewer dental malocclusions, are less likely to develop childhood leukemia, and have greater intelligence than their counterparts [13,14]. Optimal breast feeding also reduces the risks of childhood obesity, adult overweightness/obesity, and type 2 diabetes [3,15].

The timely initiation of CF at six months similarly protects against childhood illnesses and promotes health and nutritional wellbeing [16]. It has been shown by recent studies that the initiation of CF too early is associated with undernutrition [17], overweightness in children [18], as well as obesity and overweightness in later life [19], and that the delayed introduction of CF increases the risk of celiac disease in children [20], lower child growth velocity and increases childhood infections [21,22]. For these reasons, the World Health Organization (WHO) recommended EIBF followed by EBF for six months and then the introduction of CF at six months with breastfeeding [4].

However, despite the numerous short- and long-term advantages of optimal breast feeding, only 37% of children younger than six months of age are exclusively breastfed in low-income and middle-income countries [14]. Astonishingly, the EIBF percentage is less than 50% in low income countries, e.g., 42% in South Asia, 35% in West and Central Africa, and 45% in Sub-Saharan Africa [23]. Similarly, approximately one third of infants aged 4–5 months are already prematurely weaned onto solid foods, whereas about 20% of 10–11-month olds have never consumed solid foods. Furthermore, the timely introduction of complementary foods is immensely more problematic in developing countries, such as those situated in Latin America, the Caribbean, East Asia and the Pacific, where nearly half of all infants between 4 and 5 months of age are already consuming solid foods [24].

Many studies have attempted to identify factors associated with suboptimal breast feeding [25,26,27,28,29,30] and the timely initiation of complementary feeding [31,32,33,34]. Maternal socio-economic and demographic factors [26,27,35], the receipt of adequate prenatal care services, postnatal visits [25,26,29], breast related problems [30], and pre-lacteal feeds [35] have been well reported to be associated with EIBF and EBF. Similarly, maternal factors such as living in urban settlements, education, and the utilization of maternal care services (e.g., antenatal follow ups, institutional delivery, and postnatal check-ups) have been reported to predict the initiation of CF at the appropriate time [31,32]. However, such studies are sparse in Nepal.

Despite having effectively implemented the Infant and Young Child Feeding (IYCF) programme, recent studies that utilized nationally representative data from the Nepal Demographic and Population Health Survey (NDHS) in 2016 demonstrated that two thirds of Nepalese mothers exclusively breastfed children for ≤5 months, initiated breastfeeding within an hour of childbirth [36,37,38], and that 16% of mothers did not introduce complementary foods at 6–8 months. Furthermore, breast feeding and complementary feeding practices vary dramatically across Nepal [39]. Therefore, it is essential that program managers and policy makers understand local and regional child feeding practices, as well as factors affecting these practices, before establishing customized strategies aimed at improving the health and nutritional status of younger Nepalese children. The present study was undertaken to identify maternal factors associated with infant feeding practices and to determine the effects of maternal care service utilization on infant feeding practices in rural areas of Southern Nepal.

## 2. Materials and Methods

### 2.1. Study Design and Participants

Data from a cluster randomized controlled trial ‘MATRI-SUMAN’ conducted during 2015–2016 were used to identify associations between infant feeding practices, maternal factors and the utilization of maternal care services. A multistage stratified random sample of 426 pregnant women was selected from six rural village development committees (VDCs) in the Dhanusha district of Nepal. The overall response rate was 94.3%. However, of these 426 pregnant women, 379 who breastfed their child and were followed up for seven months postnatally were included in the present study. The MATRI-SUMAN trial focused on the capacity building of female community health volunteers (FCHVs) and providing information about maternal and child health care services (MCH) to pregnant women by mobile text messaging. The capacity building of FCHVs was performed by one day extensive reinforcement training to equip them with the knowledge and skills of MCH services, and mobile text messaging intervention was done through periodic mobile short messaging service (SMS) system to either pregnant woman or their family members (who could convey messages to the participant) in the intervention arm concentrating on MCH services. Details of sample selection and procedures have been found elsewhere [40].

### 2.2. Data Collection and Measures

Trial data were collected from MCH register of FCHVs and face-to-face interview by female research assistants using survey questionnaire at different time interval. The details of the follow ups and measurement can be found elsewhere [40]. We used datasets that contained information on the use of maternal care services, socio-demographic characteristics, and infant feeding practices.

### 2.3. Outcome Measures

The dependent variables of the study were: (i) early initiation of breast feeding within an hour of birth (EIBF), (ii) exclusive breastfeeding (EBF), and the (iii) initiation of complementary feeding (CF) at six months. EIBF was classified ‘yes’ if the mother initiated breast feeding within the first hour of after birth and not otherwise. EBF was recorded as ‘yes’ if a mother fed her baby only breast milk (excepting syrups and medicines). Similarly, the initiation of the CF at six months recorded as yes or no. These outcome variables were adapted from UNICEF and WHO recommendations and standards [4,41] and the measurement was based on the MCH register of FCHVs and survey questionnaires.

### 2.4. Measurement of Exposure

Two separate dimensions of exposure variables were assessed, that is, maternal care services received and sociodemographic characteristics. Maternal care services included antenatal visits, place of delivery, and postnatal care received. ANC visits were recorded as continuous variables and categorized as < or ≥4 ANC visits. Similarly, the place of delivery was categorized as either the home or health facility. PNC visits were categorized as ‘yes’ or ‘no’. The child’s sex was classified as either male or female.

### 2.5. Socio-Demographic Variables

Socio-demographic variables were adapted from previous studies. Age was categorized as: (i) <20 years, (ii) 20–34 years, or (iii) ≥35. Caste/ethnicity were classified as (i) upper caste—Brahmin, Chhetri, and Terai (Yadav, Teli, Thakur, Koiri), (ii) Adibasi/Janjati—Janjati and indigenous, and (iii) Dalit [42]. Education was recorded as years of completed education using the following three categories: (i) no education, (ii) primary—1 to 5 years of schooling, and (iii) secondary and higher ≥6 years of education [39,43]. Occupations were categorized as: (i) business, private or government, or households work; (ii) agricultural work if in own farms, and (iii) skilled or unskilled manual work. Monthly incomes were classified by tertiles: (i) 1st tertile (income < 14,333 Nepalese Rupees (Nrs)/month) (ii) 2nd tertile (Nrs 14,334–23,666/month) (iii) 3rd tertile (>Nrs. 23,666/month). Families were classified as nuclear or joint, parity as primi or multi, and birth origin as Terai or Hill on the basis of the geographical regions of Nepal [44].

### 2.6. Statistical Analyses

Infants feeding practices are reported as percentages of all infants. Chi-square tests (χ^2^) were performed to assess associations between independent variables with EIBF, EBF for six months, and the initiation of CF at six months. Variables found to be significant by Chi-square testing were subjected to multivariate logistic regression analysis. Independent variables with a p-value of <0.1 were entered into the multivariate analysis. Unadjusted and adjusted odds ratios with 95% confidence intervals (CIs) are reported. The statistical analysis was conducted using SPSS ver. 21.0 (SPSS, IBM, Armonk, NY, USA).

### 2.7. Ethics

Ethical approval for the ‘MATRI-SUMAN’ protocol was obtained from the Nepal Health Research Council, Nepal (approval no: 101) and the ethics committee of the Institute of Medical Sciences, Banaras Hindu University, India (approval no: ECR/526/Inst/UP/2014 Dt.31.1.14), and the District Public Health Office, Dhanusha, Nepal (Ref. 2245). Additional ethical approval was also received from the Institutional Review Board of Janaki Medical College for the data analysis. The aims and objectives of the study were explained to all study subjects, who provided written informed consent. All personal identifiers were removed before the data analysis.

## 3. Results

### 3.1. Infant Feeding Practices of Mothers

Infant feeding practices of among mothers in rural Terai are detailed in Table 1. For the 379 study subjects, feeding practices were as follows; EIBF 41.4%, EBF 53.0%, and 43% initiated CF at six months.

### 3.2. Maternal Characteristics and Utilization of Maternal Care Services

The majority of the study subjects (69.9%) were 20–34 years old, 62.5% were from an upper caste group, 71.8% were Terai by birth, 23.5% had no education, 83.1% worked in the agricultural or service/business/household sectors, 65.7% had family incomes in the second or third tertile, slightly more than half (52.5%) were from joint families, 60.9% were multiparous, and 52.3% of the babies were female. Maternal factors such as birth origin, education, occupation, family income, and family type were significantly associated with EIBF; caste/ethnicity, education, occupation, family income, and parity were significantly associated with EBF; and birth origin, education, occupation, and child sex were significantly associated with the initiation of CF at six months (Table 2).

The utilization of maternal care services associated with infant feeding practices is summarized in Table 3. Half (50.9%) of the study subjects received MATRI-SUMAN intervention, 60.9% visited an ANC more than four times, 58.0% of childbirths were at home, and 57.8% visited a PNC. All variables associated with the utilization of maternal care services were significantly associated with the three child feeding practices.

### 3.3. Maternal Factors and the Utilization of Maternal Care Services Associated with Infant Feeding Practices

Maternal factors and the utilization of maternal care services associated with infant feeding practices with unadjusted odds ratios are provided in Table 4. A number of maternal factors and maternal care service utilization factors, that are caste/ethnicity, birth origin, education, occupation, family income, family type, MATRI-SUMAN intervention, ANC visits, place of delivery, and PNC visits, were significantly associated with EIBF; education, occupation, family income, parity, MATRI-SUMAN intervention, ANC visits, place of delivery, and PNC visits were significantly associated with EBF; and birth origin, education, occupation, family income, child sex, MATRI-SUMAN intervention, ANC visits, place of delivery, and PNC visits were significantly associated with the initiation of CF at six months by univariate analyses.

Multiple logistic regression model results with adjusted odds ratio are summarized in Table 5. Maternal education secondary and higher (aOR 2.2; 95% CI (1.2–4.2)), occupation in the service/business/household sectors (aOR 2.2; 95% CI (1.2–4.2)), receipt of MATRI-SUMAN intervention (aOR 1.7; 95% CI (1.1–2.9)), ≥4 ANC visits (aOR 3.2; 95% CI (1.2–8.0)), and delivery in a health facility (aOR 1.9; 95% CI (1.0–3.4)) had higher odds ratios for EIBF.

Mothers with a primary level of education (aOR 2.2; 95% CI (1.1–4.4)), a secondary or higher level of education (aOR 5.5; 95% CI (1.8–16.1)), working in the service/business/household sectors (aOR 2.1; 95% CI (1.1–4.1)), primipara (aOR 2.2; 95% CI (1.2–4.0)), that received MATRI-SUMAN intervention (aOR 1.8; 95% CI (1.1–3.2)), that had visited an ANC ≥ 4 times (aOR 3.1; 95% CI (1.3–7.5)) and visited a PNC (aOR 2.4; 95% CI (1.1–5.6)) had higher odds ratios of EBF. However, mothers aged 35–45 years were less likely (aOR 0.3; 95% CI (0.1–0.7)) to have performed EBF than their counterparts.

Mothers with a secondary or higher education level (aOR 2.2; 95% CI (1.2–3.9)), male baby (aOR 1.7; 95% CI (1.1–2.7)), received MATRI-SUMAN intervention (aOR 1.6; 95% CI (1.0–2.7)), and visited a PNC (aOR 2.3; 95% CI (1.0–5.1)) had higher odds ratios for initiating CF at six months.

## 4. Discussion

In the study, 41.4%, 53%, and 43% of 379 ever-breastfed mothers initiated breast feeding within an hour of the birth (EIBF), practiced exclusive breast feeding (EBF), and initiated complementary feeding (CF) at six months, respectively, and these percentages are lower compared with those in a recent Nepalese Demographic and population health survey report 2016 [39]. However, the proportion of mothers that performed EIBF in the present study is similar to those reported in other developing countries like Brazil [45] and Ethiopia [46], but higher than that found in an Indian study [47]. Likewise, the prevalence of EBF in Nigerian [48] and Ethiopian [49] studies was reported to be 37.3% and 54.5%, respectively, values which are substantially higher than that found in a study conducted in Indian urban slums, in which only 7.8% were exclusively breast fed [50]. However, a recent Nepalese study conducted using nationally representative data also reported an EBF rate for children of ≤five months was 66.3%, which is higher than that found in the present study. These differences were probably due to different study population sizes and settings, for example, some used nationally representative datasets [36,38], whereas our study was conducted in the rural Southern Terai. In addition, different levels of education, wealth quintiles, and existing local traditions and beliefs might have impacted results.

In the present study, only slightly more than two fifths of mothers initiated CF at 6 months, which is lower than those found in some Ethiopian studies performed in different parts of the country [32,33,51], which reported >60% of mothers initiated CF at six months. Studies conducted in other developing countries like Ghana [52], Ethiopia [53], and Bangladesh [54], also presented higher percentages of mothers initiated CF at six months. Recent local Nepalese studies [55,56] also reported 57% of mothers initiated CF at six months, and these values are also higher than those found in the present study. This wide variation in the prevalence of the timely initiation of CF may have been due to the fact that the present study was conducted in rural areas of the South-East region of Nepal, whereas the two other Nepalese studies were conducted in Western Nepal. Differences between Nepalese studies may also have been influenced by different sample sizes, literacy statuses, and methodological differences. Importantly, we dichotomized mothers based on whether they initiated CF at exactly six months, and several other studies used a range of 6–8 months [52,54,57], which might also have contributed to reported differences.

We found maternal education and the receipt of MATRI-SUMAN intervention increased the odds ratio of initiating EIBF and the practice of EBF and CF. Several other reports concur regarding a positive association between maternal education and EIBF, EBF and CF [28,30,36,53,58,59], presumably because educated women are more aware about healthy and timely child feeding practices. In fact, maternal education has been reported to be associated with improved child feeding practices [60]. MATRI-SUMAN intervention on the other hand might have influenced rural women to adopt recommended child feeding practices. MATRI-SUMAN intervention has two components, namely the training of female community health volunteers and mobile text messaging directed at pregnant and post-natal women; both approaches were directed at increasing the utilization of maternal and child health care services, and thereby, improved maternal and child health. The MATRI-SUMAN intervention has been previously described in detail [40].

In addition, we observed that a maternal occupation in the service/business/household sectors resulted in higher odds ratios for EIBF and EBF than those observed in the manual labor sector. This finding agrees with some previous reports [25,28,61], but conflicts with others [62]. Alzaheb et al. and Adugna et al. reported working mothers were less likely to adopt EBF [25,61], and a review paper issued by Alzaheb et al. showed working mothers adopted EIBF less frequently than non-working mothers [28]. However, in rural areas of Southern Terai, small businesses are conducted in homes or nearby, and we categorized these mothers as working at home, whereas women engaged in manual labor usually work long hours far away from their homes, and thus find breastfeeding more difficult. We also found older mothers (35–45 years old) were less likely to practice EBF. The relation between maternal age and EBF is unclear. For example, a recent study concluded that younger women were less likely to practice EBF [63], whereas others support the notion that an older maternal age predicts the use of non-exclusive breast feeding [47,48]. In addition, a study performed in Nigeria discovered that the practice of EBF increases with a mother’s age to a peak at around 32 years, and that teenage mothers and mothers older than 32 years tended not to practice EBF [64]. It is possible older women have more roles and responsibilities in addition to child care and that they are required to look after other family members, respectively in poor rural areas. On the other hand, several studies have reported primiparity as a potential risk factor for not practicing EBF [65,66,67,68], whereas we found primiparous mothers were more likely to practice EBF than multiparous mothers. Reports differ regarding the association between parity and EBF. Studies conducted in China and Sweden found no association between the two [69,70], whereas others [71,72] reported primiparous mothers were more likely to adopt EBF, which highlights the importance of educating multiparous women. Furthermore, as has been reported previously [51,73], we found CF feeding at six months was more common for male babies, which may reflect the long-standing traditional gender norm that discriminates against female child feeding [51].

Importantly, our study confirms that some specific components of maternal care services, such as antenatal visits, childbirth at a health facility, and postnatal visits, significantly impact child feeding practices. Mothers that visited ANCs four or more times were found to be significantly more likely to practice both EIBF and EBF, and childbirth at a health facility was significantly associated with EIBF. In addition, PNC visits appeared to increase the likelihood of EBF and the initiation of CF at six months. A number of studies have demonstrated a significant positive association between prenatal care visits and EIBF [33,35,46,74] and EBF [26,50]. Similarly, we found a higher odds ratio of EIBF among mothers that gave birth at a health facility, which concurs with reports issued in other developing countries [25,48]. In particular, an Ethiopian and a Nepalese study identified that institutional delivery increased EIBF [32,36], and another Nepalese study found home delivery was associated with EBF [37]. Nonetheless, a more recent study conducted in Nigeria concluded that non-exclusive breast feeding was attributable to no antenatal care visits, home delivery, and delivery assisted by a non-health professional, which is in line with our study findings [75]. These findings may have arisen because maternal exposure to breast feeding counseling by health workers during childbirth increased maternal knowledge about breastfeeding practices. Furthermore, maternal exposure to health education classes on breast feeding and an adequate knowledge of breast feeding improves breast feeding practices [76]. Thus, institutional child birth increases the likelihood of mothers adopting EIBF and EBF [29]. These observations suggest modifiable factors, such as the promotion of childbirth at health facility and education about breast feeding, which can present excellent means of promoting breast feeding practices in low income countries like Nepal. In addition, an Ethiopian study also showed that postnatal visits had a significant positive association with the timely initiation of complementary feeding [31], and in other studies, childbirth at a health institution were found to predict timely complementary feeding [32,77]. We recommend that interventions to promote complementary feeding at recommended times in health care settings should be promptly implemented.

Our study has several valuable strengths. First, the response rate was very high. Second, we assessed the effects of a number of maternal and maternal care service factors on child feeding practices using three dependent variables, that is, EIBF, EBF, and timely initiation of CF using the WHO recommendation and indicators in the same cohort who were involved in the trial. Third, our study made use of a dataset obtained at different time intervals from the same cohort, which may provide reliable information. However, the study has some specific limitations that should be understood. First, our measurements were based on the MCH register as well as maternal recall at different time intervals of the same cohort, and thus may have been influenced by measurement and recall biases. Second, we only assessed whether mothers started complementary feeding at six months and our study could not indicate the appropriateness of the first CF in terms of dietary energy. Third, since the MATRI-SUMAN trial ended follow up visits at seven months after the birth of the baby, our methods could not assess other potentially important indicators of complementary feeding practices, such as meal frequency, dietary diversity, and minimum acceptable diet. Fourth, like many other cross-sectional studies, the present study does not address causality.

## 5. Conclusions

In the present study, reported rates of EIBF, EBF, and CF were found to be lower than those reported in other developing countries. Maternal education and the receipt of MATRI-SUMAN intervention positively impacted EIBF, EBF, and CF, and maternal occupation in the service/business/household sector positively influenced EIBF and EBF. Primiparity was found to be protective, but an older age (35–45 years) negatively influenced EBF. In addition, it was observed that male babies were more likely to receive complementary feeding at the recommended time. The utilization of maternal care services, such as ≥4 antenatal care visits significantly and positively promoted the practice of EIBF and EBF, whereas childbirth at a health facility was significantly associated with EIBF alone. In addition, a PNC visit was found to significantly increase the probability of EBF and of initiating CF at six months. Thus, we recommend that the utilization of maternal care services, such as ANCs, the use of health institutions for child birth, and postnatal visits should be recommended to improve child feeding practices, and that maternal factors which impact child feeding practices be carefully considered when designing strategies and interventions. In addition, we suggest a community-based study be undertaken to improve the understanding of the effects of sub-optimal breast feeding and complementary feeding among Nepalese infants.

## Figures and Tables

**Table 1 ijerph-16-01887-t001:** Infant feeding practice among mothers in rural Southern Nepal, 2016.

Infant Feeding Practices	Yes, *n* (%)	No, *n* (%)	Total, *N* (%)
Initiation of breast feeding within first hour of birth	157 (41.4)	222 (58.6)	379 (100)
Exclusive breast feeding for 6 months	201 (53.0)	178 (47.0)	379 (100)
Initiation of complementary feeding at 6 months	163 (43.0)	216 (57.0)	379 (100)

**Table 2 ijerph-16-01887-t002:** Association between maternal characteristics of mothers and infant feeding practices in rural Southern Nepal, 2016.

Variables	Total, *N* (%) 379 (100)	Infant Feeding Practices, Yes, *n* (%)		
		Initiation of Breast Feeding within First Hour of Birth Yes, n (%) 157 (41.4)	Exclusive Breast Feeding for 6 Months Yes, *n* (%) 201 (53.0)	Initiation of Complementary Feeding at 6 Months Yes, *n* (%) 163 (43.0)
Age (years)		*p* = 0.371	*p* = 0.545	*p* = 0.127
<20	86 (22.7)	41 (47.7)	50 (58.1)	44 (51.2)
20–34	265 (69.9)	106 (40.0)	136 (51.3)	105 (39.6)
≥35 years	28 (7.4)	10 (35.7)	15 (53.6)	14 (50.0)
Caste/ethnicity		*p* = 0.141	*p* < 0.001	*p* = 0.296
Dalit	58 (15.3)	18 (31.0)	132 (55.7)	99 (41.8)
Adibasi/Janajati	84 (22.2)	40 (47.6)	51 (60.7)	42 (50.0)
Upper caste Group	237 (62.5)	99 (41.8)	18 (31.0)	22 (37.9)
Birth origin		*p* = 0.025	*p* = 0.097	*p* = 0.038
Hill	107 (28.2)	54 (50.5)	64 (59.8)	108 (39.7)
Terai	272 (71.8)	103 (37.9)	137 (50.4)	55 (48.6)
Women education		*p* < 0.001	*p* < 0.001	*p* < 0.001
No education	89 (23.5)	23 (25.8)	16 (18.0)	22 (24.7)
Primary	135 (35.6)	42 (31.1)	69 (51.1)	48 (35.6)
Secondary & higher	155 (40.9)	92 (59.4)	116 (74.8)	93 (60.0)
Women occupation		*p* < 0.0001	*p* < 0.001	*p* < 0.001
Labor	64 (16.9)	17 (26.6)	13 (20.3)	16 (25.0)
Agricultural work	122 (32.2)	36 (29.5)	55 (45.1)	47 (38.5)
Service/business/ household works	193 (50.9)	104 (53.9)	133 (68.9)	100 (51.8)
Family income		*p* = 0.006	*p* = 0.002	*p* = 0.125
1st tertile	130 (34.3)	40 (30.8)	53 (40.8.)	47 (36.2)
2nd tertile	121 (31.9)	53 (43.8)	68 (56.2.)	54 (44.6)
3rd tertile	128 (33.8)	64 (50.0)	80 (62.5)	62 (48.8)
Types of family		*p* = 0.029	*p* = 0.260	*p* = 0.341
Nuclear	180 (47.5)	85 (47.2)	90 (50.0.)	82 (45.6)
Joint	199 (52.5)	72 (36.2)	111 (55.8)	81 (40.7)
Parity		*p* = 0.430	*p* < 0.001	*p* = 0.177
Primi	148 (39.1)	65 (43.9)	95 (64.2)	70 (47.3)
Multi	231 (60.9)	92 (39.8)	106 (45.9)	93 (40.3)
Sex of Child		*p* = 0.05	*p* = 0.097	*p* = 0.004
Male	178 (47.7)	77 (43.3)	102 (57.3)	90 (50.6)
Female	195 (52.3)	79 (40.5)	95 (48.7)	70 (35.9)

**Table 3 ijerph-16-01887-t003:** Association between utilization of maternal care services and infant feeding practices in rural Southern Nepal, 2016.

Variables	Total, N (%) 379 (100)	Infant Feeding Practices (Yes, %)		
		Initiation of Breast Feeding First Hour of Birth Yes, *n* (%) 157 (41.4)	Exclusive Breast Feeding for 6 Months Yes, *n* (%)201 (53.0)	Initiation of Complementary Feeding at 6 Months Yes, *n* (%) 163 (43.0)
MCH intervention (MATRI-SUMAN)		*p* = 0.002	*p* = 0.001	*p* = 0.002
Received	193 (50.9)	95 (49.2)	118 (61.1)	98 (50.8)
Not received	186 (49.1)	62 (33.3)	83 (44.6)	65 (34.9)
Number of ANC visit		*p* < 0.001	*p* < 0.001	*p* < 0.001
<4 ANC	148 (39.1)	30 (20.3)	31 (29.9)	35 (23.6)
4 or more	231 (60.9)	127 (55.0)	170 (73.6)	128 (55.4)
Place of Delivery		*p* < 0.001	*p* < 0.001	*p* = 0.004
Home	220 (58.0)	61 (27.7)	88 (40.0)	73 (33.2)
Health facility	159 (42.0)	96 (60.4)	113 (71.1)	90 (56.6)
PNC visit		*p* < 0.0001	*p* = 0.01	*p* < 0.001
Yes	219 (57.8)	120 (54.8)	162 (74.0)	39 (24.4)
No	160 (42.2)	37 (23.1)	39 (24.4)	124 (56.6)

MCH, maternal and child health; MATRI-SUMAN, Maternal Alliance for Technological Research Initiative on Service Utilization and Maternal Nutrition; ANC, antenatal care; PNC, postnatal care.

**Table 4 ijerph-16-01887-t004:** Maternal factors and utilization of maternal care services associated with infant feeding practice in rural Southern Nepal, 2016 by Unadjusted Odds Ratio (OR).

Variables	Infant Feeding Practices (OR, 95% CI)		
	Initiation of Breast Feeding within First Hour of Birth	Exclusive Breast Feeding for 6 Months	Initiation of Complementary Feeding at 6 Months
Age (Years)			
<20	1.00	1.00	1.00
20–34	0.7 (0.4–1.1)	0.7 (0.4–1.2)	0.6 (0.3–1.0)
35–45	0.6 (0.2–1.4)	0.8 (0.3–1.9)	0.9 (0.4–2.2)
Caste/ethnicity			
Dalit	1.00	1.00	1.00
Adibasi/Janajati	1.5 (0.8–2.9)	1.2 (0.7–2.0)	1.6 (0.8–3.2)
Upper caste group	2.0 (1.1–4.0)	0.3 (0.1–0.6)	1.1 (0.6–2.1)
Birth origin			
Terai	1.00	1.00	1.00
Hill	1.6 (1.1–2.6)	1.4 (0.9–2.3)	1.6 (1.0–2.5)
Women education			
No education	1.00	1.00	1.00
Primary	1.2 (0.7–2.3)	4.7 (2.5–9.0)	1.6 (0.9–3.0)
Secondary & higher	4.1 (2.3–7.4)	13.5 (7.0–26.0)	4.5 (2.5–8.1)
Women occupation			
Labor	1.00	1.00	1.00
Agricultural work	1.1 (0.5–2.2)	3.2 (1.5–6.5)	1.8 (0.9–3.6)
Service/business/ household works	3.2 (1.7–6.0)	8.6 (4.4–17.1)	3.2 (1.7–6.0)
Family Income			
1st tertile	1.00	1.00	1.00
2nd tertile	1.7 (1.1–2.9)	1.8 (1.1–3.0)	1.4 (0.8–2.3)
3rd tertile	2.2 (1.3–3.7)	2.4 (1.4–3.9)	1.6 (1.0–2.7)
Types of family			
Joint	1.00	1.00	1.00
Nuclear	1.5 (1.0–2.3)	1.2 (0.8–1.8)	0.8 (0.5–1.2)
Parity			
Multi	1.00	1.00	1.00
Primi	1.8 (0.7–1.7)	2.1 (1.3–3.2)	1.3 (0.8–2.0)
Sex of Child			
Female	1.00	1.00	1.00
Male	1.1 (0.7–1.6)	1.4 (0.9–2.1)	1.8 (1.2–2.7)
MCH Intervention (MATRI-SUMAN)			
Not received	1.00	1.00	1.00
Received	1.9 (1.2–2.9)	2.0 (1.3–3.1)	1.9 (1.2–2.9)
ANC visit			
<4 ANC	1.00	1.00	1.00
4 or more	4.8 (2.9–7.2)	10 (6.4–17.2)	4.0 (2.5–6.3)
Place of delivery			
Home	1.00	1.00	1.00
Health facility	3.9 (2.5–6.1)	3.6 (2.3–5.7)	2.6 (1.7–4.0)
PNC visit			
No	1.00	1.00	1.00
Yes	4.0 (2.5–6.3)	8.8 (5.5–14.1)	4.0 (2.5–6.3)

MCH, maternal and child health; MATRI-SUMAN, Maternal Alliance for Technological Research Initiative on Service Utilization and Maternal Nutrition; ANC, antenatal care; PNC, postnatal care.

**Table 5 ijerph-16-01887-t005:** Adjusted Odds Ratio (aOR) for infant feeding practices among rural mothers of Southern Nepal, 2016.

Variable	Infant Feeding Practices (aOR, 95% CI)		
	Initiation of Breast Feeding First Hour of Birth	Exclusive Breast Feeding for 6 Months	Initiation of Complementary Feeding at 6 Months
Age (Years)			
<20	1.00	1.00	1.00
20–34	0.4 (0.1–1.2)	0.3 (0.1–1.1)	0.6 (0.2–1.7)
35–45	0.5 (0.2–1.7)	0.3 (0.1–0.7)	0.9 (0.4–1.6)
Caste/ethnicity			
Dalit	1.00	1.00	1.00
Adibasi/Janajati	1.7 (0.5–5.1)	1.1 (0.4–2.7)	1.0 (0.4–2.1)
Upper caste group	0.7 (0.4–1.4)	0.6 (0.3–1.2)	1.1 (0.6–1.9)
Birth origin			
Terai	1.00	1.00	1.00
Hill	1.4 (0.7–2.5)	1.1 (0.3–3.0)	1.1 (0.6–1.9)
Women education			
No education	1.00	1.00	1.00
Primary	1.2 (0.4–3.3)	2.2 (1.1–4.4)	2.2 (0.8–5.6)
Secondary & higher	2.2 (1.2–4.2)	5.5 (1.8–16.1)	2.2 (1.2–3.9)
Women occupation			
Labor	1.00	1.00	1.00
Agricultural work	1.3 (0.4–3.8)	1.5 (0.5–5.0)	1.1 (0.6–3.0)
Service/business/household works	2.2 (1.2–4.2)	2.1 (1.1–4.1)	1.4 (0.5–4.0)
Family Income			
1st tertile	1.00	1.00	1.00
2nd tertile	1.0 (0.5–2.0)	1.2 (0.6–2.4)	1.0 (0.4–2.4)
3rd tertile	1.1 (0.6–2.1)	1.5 (0.7–3.2)	1.1 (0.5–2.0)
Types of family			
Joint	1.00	1.00	1.00
Nuclear	1.3 (0.8–2.1)	1.6 (0.9–2.9)	1.0 (0.6–1.6)
Parity			
Multi	1.00	1.00	1.00
Primi	1.2 (0.6–2.1)	2.2 (1.2–4.0)	1.3 (0.8–2.0)
Sex of child			
Female	1.00	1.00	1.00
Male	1.1 (0.6–1.7)	1.2 (0.7–2.0)	1.7 (1.1–2.7)
MCH Intervention (MATRI-SUMAN)			
Not received	1.00	1.00	1.00
Received	1.7 (1.1–2.9)	1.8 (1.1–3.2)	1.6 (1.0–2.7)
ANC visit			
<4 ANC	1.00	1.00	1.00
4 or more	3.2 (1.2–8.0)	3.1 (1.3–7.5)	1.2 (0.5–2.8)
Place of delivery			
Home	1.00	1.00	1.00
Health facility	1.9 (1.0–3.4)	1.2 (0.6–2.4)	1.1 (0.6–2.0)
PNC visit			
No	1.00	1.00	1.00
Yes	1.0 (0.4–2.3)	2.4 (1.1–5.6)	2.3 (1.0–5.1)

MCH, maternal and child health; MATRI-SUMAN, Maternal Alliance for Technological Research Initiative on Service Utilization and Maternal Nutrition; ANC, antenatal care; PNC, postnatal care. Variables entered for each outcome variables: age group, caste/ethnicity, birth origin, women’s occupation, women’s education, family income, types of family, MCH intervention received, parity, ANC visit, place of delivery, PNC, sex of child.

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
