# Peer review of "Maternal Factors and the Utilization of Maternal Care Services Associated with Infant Feeding Practices among Mothers in Rural Southern Nepal"

_ijerph, 2019, doi:10.3390/ijerph16111887_

Round 1
Reviewer 1 Report
Is interesting that you have analyzed feeding practices in infants and factors associated in rural area. I have few comments and suggestions for your consideration:
I suggest adding in the introduction the growth risk associated with late introduction of CF.
Please review the references numbers, I noticed some mistakes, could be that you added one reference and did not correct the numbers.
I suggest describing the indicators as mentioned in the WHO references (example: early intiation of breastfeeding means initiation within one hour; use one or the other)
The questions usually should have a couple of elements, for example: asking "did you breastfed your baby within one hour?" is a leading question.. if that was the way it was asked acknowledge the possibility of bias
In the section results might be good to include results related with mothers not showing the appropriate behaviour.
Review numbers/percentages in your tables (does not add to 100%)
Antenatal care usually does not affect initiation of complementary foods as that is not a subject of counselling or orientation at that time (more focus on early initiation and exclusive breastfeeding in relation to infant feeding practices)
Postnatal care visit should not be considered when analyzing early initiation
Author Response
First reviewer: first round
Journal: International Journal Environmental Research and Public Health
Manuscript ID: ijerph-510933
Title: Maternal factors and the utilization of maternal care services associated with infant feeding practices among mothers in rural southern Nepal
Comments and Suggestions for Authors
General comments
Is interesting that you have analyzed feeding practices in infants and factors associated in rural area. I have few comments and suggestions for your consideration:
Response: Agree. We thank you very much and highly appreciate reviewer’s valuable comments and suggestions that are directed to improve our paper. We have revised the manuscript based on reviewers’ comments and suggestions and marked all changes are marked with blue colored writing in this revised version of the manuscript to allow reviewers’ verifications.
Comment: I suggest adding in the introduction the growth risk associated with late introduction of CF.
Response: Agree. Added as suggested.
Comment: Please review the references numbers, I noticed some mistakes, could be that you added one reference and did not correct the numbers.
Response: Agree. All cited references are checked and corrected.
Comment: I suggest describing the indicators as mentioned in the WHO references (example: early initiation of breastfeeding means initiation within one hour; use one or the other)
Response: Partially agree. It has been mentioned in the method section with citation.
Comment: The questions usually should have a couple of elements, for example: asking "did you breastfed your baby within one hour?" is a leading question. If that was the way it was asked acknowledge the possibility of bias.
Response: Agree. This has been changed with clear information of Matri-Suman Trial with reference.
Comment: In the section results might be good to include results related with mothers not showing the appropriate behavior.
Response: Agree. Thank very much for the suggestion. Unfortunately we did not collect this information in this study; as the result of which we are unable to present this information in this paper.
Comment: Review numbers/percentages in your tables (does not add to 100%)
Response: Agree. Get corrected.
Comment: Antenatal care usually does not affect initiation of complementary foods as that is not a subject of counseling or orientation at that time (more focus on early initiation and exclusive breastfeeding in relation to infant feeding practices).
Response: Agree. Revised.
Comment: Postnatal care visit should not be considered when analyzing early initiation.
Response: Agree. Revised.
Reviewer 2 Report
General comments to the authors
This paper describes a study investigating the infant feeding practices of rural mother in Nepal and their association with a care services. Some further clarifications are required.
Introduction
Page 2, line 2 – what is %
In the discussion of breastfeeding, the role of BF in infant growth and infection is well established. However, the link between leukaemia, IQ and dental health are not well established, make this clear. IQ is most likely confounded by maternal IQ.
Regarding timely CF. Obesity is not the only problem, too early or late can also lead to growth restriction. Either due to too early introduction of low energy foods or due to breast milk being or poor source of Fe and Zn after 6 months. Also too early CF can lead to increased infections
“Furthermore the timely introduction of CF…” this sentence needs to be expanded upon, what is furthermore to the previous discussion? Also there in no reference 22 in the reference list.
Methods
When were the women interviewed, how likely were they to remember the specific timing of feeding at birth and introducing CF. How was the question worded and how well did the women understand the question about initiation of CF at 6 months and how strictly is this applied. Many women would not remember to the actual day – perhaps within weeks. Also do you consider 5 months and 3 weeks too early? Yes the WHO recommendation is 6 months, but there is not a magic number – what was your tolerance.
Can you further define Terai or Hill.
Did all the women have phones to receive text messages. If not how many of your women had mobile phones to receive the text messages? Having a phone could have had an effect compared to those who did not.
Could you give a brief overview of the Matri-Suman intervention. I know you have previously published this, but to give background so the paper can be read stand alone.
Discussion
You state here you looked at initiation of CF at exactly 6 months – so do you mean on the date the child was 6 months old – was there no tolerance? What if the infant was 5 months and 30 days? Or 6 months plus one week. How well do you think the woman remembered this information?
Likewise BF within one hour of birth – birthing can be quite chaotic – so it can be difficult for the mother to remember the specifics. Also if she requires some intervention, caesarean or even suturing this can delay feeding, but this is not due to lack of knowledge.
Another limitation is you do not know the appropriateness of the first CF in terms of dietary energy
Minor comments
Use the term tertile not tercile.
Author Response
Second reviewer: first round
Journal: International Journal Environmental Research and Public Health
Manuscript ID: ijerph-510933
Title: Maternal factors and the utilization of maternal care services associated with infant feeding practices among mothers in rural southern Nepal
Comments and Suggestions for Authors
General comments to the authors
This paper describes a study investigating the infant feeding practices of rural mother in Nepal and their association with a care services. Some further clarifications are required.
Response: Agree. Thank you very much for the specific comments and suggestions. We have revised the manuscript on the basis of comments and suggestions and all changes are marked with blue colored writing in this revised version of the manuscript to allow reviewers’ verifications.
Introduction
Comments: Page 2, line 2 – what is %
Response: Agree. Sentence restructured.
Comments: In the discussion of breastfeeding, the role of BF in infant growth and infection is well established. However, the link between leukaemia, IQ and dental health are not well established, make this clear. IQ is most likely confounded by maternal IQ.
Response: Agree. The same sentence has been restructured.
Comments: Regarding timely CF. Obesity is not the only problem, too early or late can also lead to growth restriction. Either due to too early introduction of low energy foods or due to breast milk being or poor source of Fe and Zn after 6 months. Also too early CF can lead to increased infections.
Response: Agree. Problem of growth faltering and infections has been added.
Comments: “Furthermore the timely introduction of CF…” this sentence needs to be expanded upon, what is furthermore to the previous discussion? Also there in no reference 22 in the reference list.
Response: Agree. The sentence has been made clear in this revised version of the manuscript and also all cited references are checked and corrected.
Methods
Comments: When were the women interviewed, how likely were they to remember the specific timing of feeding at birth and introducing CF. How was the question worded and how well did the women understand the question about initiation of CF at 6 months and how strictly is this applied. Many women would not remember to the actual day – perhaps within weeks. Also do you consider 5 months and 3 weeks too early? Yes the WHO recommendation is 6 months, but there is not a magic number – what was your tolerance.
Response: Agree. Record was validated with the maternal and child health register. The current version of the manuscript has addressed the issue with clear description of about Matri-Suman Trial.
Comments: Can you further define Terai or Hill.
Response: Agree. This has been defined with citation. For your kind information, Nepal has been divided into three ecological regions based on altitude: mountains (the northern Himalayan rage, mostly covered with snow; more than 3000–8850 meters), hills (middle section of hills; 600–3,000 meters altitude), and Terai (the southern plain area; below 600 meters).
Comments: Did all the women have phones to receive text messages. If not how many of your women had mobile phones to receive the text messages? Having a phone could have had an effect compared to those who did not.
Response: Agree. Either pregnant woman, her partner or near relative had mobile phone to access text message in the intervention area.
Comments: Could you give a brief overview of the Matri-Suman intervention. I know you have previously published this, but to give background so the paper can be read stand alone.
Response: Agree. We have now written the overview of Matri-Suman intervention as suggested.
Discussion
Comments: You state here you looked at initiation of CF at exactly 6 months – so do you mean on the date the child was 6 months old – was there no tolerance? What if the infant was 5 months and 30 days? Or 6 months plus one week. How well do you think the woman remembered this information?
Response: Agree. Revised version of the manuscript has addressed this methodological issue.
Comments: Likewise BF within one hour of birth – birthing can be quite chaotic – so it can be difficult for the mother to remember the specifics. Also if she requires some intervention, caesarean or even suturing this can delay feeding, but this is not due to lack of knowledge.
Response: Agree. The issue of recall bias has been added in the limitation of the study.
Comments: Another limitation is you do not know the appropriateness of the first CF in terms of dietary energy
Response: Agree. Addressed it as one the limitations of the study.
Minor comments
Comments: Use the term tertile not tercile.
Response: Agree. Corrected.